# Rats that learn to vocalize for food reward emit longer and louder appetitive calls and fewer short aversive calls

**Agnieszka D. Wardak[1], Krzysztof H. Olszyński[1], Rafał Polowy[1], Jan Matysiak[2], Robert K. Filipkowski**[1] *

**1** Behavior and Metabolism Research Laboratory, Mossakowski Medical Research Institute, Polish Academy of Sciences, Warsaw, Poland, **2** Institute of Psychology, University of Economics and Human Sciences in Warsaw, Warsaw, Poland

* rfilipkowski@imdik.pan.pl

**Data Availability Statement:** The data that support the findings of this study are openly available in Mendeley Data. https://data.mendeley.com/datasets/brkjcrhzsv/1.

## Abstract

Rats are social animals that use ultrasonic vocalizations (USV) in their intraspecific communication. Several types of USV have been previously described, e.g., appetitive 50-kHz USV and aversive short 22-kHz USV. It is not fully understood which aspects of the USV repertoire play important functions during rat ultrasonic exchange. Here, we investigated features of USV emitted by rats trained in operant conditioning, is a form of associative learning between behavior and its consequences, to reinforce the production/emission of 50-kHz USV. Twenty percent of the trained rats learned to vocalize to receive a reward according to an arbitrarily set criterion, i.e., reaching the maximum number of proper responses by the end of each of the last three USV-training sessions, as well as according to a set of measurements independent from the criterion (e.g., shortening of training sessions). Over the training days, these rats also exhibited: an increasing percentage of rewarded 50-kHz calls, lengthening and amplitude-increasing of 50-kHz calls, and decreasing number of short 22-kHz calls. As a result, the potentially learning rats, when compared to non-learning rats, displayed shorter training sessions and different USV structure, i.e. higher call rates, more rewarded 50-kHz calls, longer and louder 50-kHz calls and fewer short 22-kHz calls. Finally, we reviewed the current literature knowledge regarding different lengths of 50-kHz calls in different behavioral contexts, the potential function of short 22-kHz calls as well as speculate that USV may not easily become an operant response due to their primary biological role, i.e., communication of emotional state between conspecifics.

## Introduction

Adult rats use two main types of ultrasonic vocalizations (USV) to communicate: 22 kHz and 50 kHz. 22-kHz USV, known as alarm calls, are emitted mostly in aversive situations and are connected with a negative state [1]. 50-kHz USV are emitted in an appetitive or social context and are connected with a positive state [2]. These two main call types are significantly different in acoustic parameters: 22-kHz USV have a low sound frequency between 18–32 kHz and are

**Funding:** This work was funded by the National Science Centre, Poland, grant OPUS no. 2015/19/B/NZ4/03393 and by Mossakowski Medical Research Institute Research Fund, grant no. FBW-17. The funders had no role in study design, data collection and analysis, decision to publish, or preparation of the manuscript.

**Competing interests:** The authors have declared that no competing interests exist.

divided into long (>300 ms) and short (<300 ms) calls [3]; 50 kHz USV are higher in frequency, between 32–96 kHz, and are shorter in duration (30–50 ms) [4]. Recently, we have also described a new type of rat aversive USV, namely 44-kHz USV, of frequency higher than 32 kHz and duration longer than 150 ms emitted in rats experiencing intense stress [5].

Operant (or instrumental) conditioning is a form of associative learning between a behavior and its consequences which determines the probability of this behavior reoccurring [6]. Operant/instrumental reflexes were first described by Konorski and Miller as "type II conditioned reflexes" [7]; for English version [see 8]. To affect the behavior change, operant conditioning requires a punishment or reinforcement (positive or negative) to influence the likelihood and incidence of a particular behavior in the future. In the case of rats, operant conditioning classically takes place in a laboratory apparatus called a *Skinner box* [9] equipped with levers to press or holes to perform nosepokes to get a reward or to avoid punishment. We wanted to condition the rats to form another response apart from nosepoke or lever-pressing, i.e., emission of USV.

There have been several studies using USV in operant conditioning. USV-playback was used to evoke emotional contagion and to change the affective states of receiver rats. Rats exposed to 50-kHz USV responded to an ambiguous cue as positive, and as negative when exposed to 22-kHz USV [10]. Also, rats were able to learn operant procedures to discriminate between 50-kHz and 22-kHz USV, which allowed to identify key USV features necessary for the discrimination [11]. Finally, USV were used in self-administration experiments. One of them showed that rats preferred to press the lever to 50-kHz-playback from a stranger rat over familiar one [12].

To our knowledge, there is only one published operant conditioning protocol that rewards rats for USV emission, which is based on initially exposing a male rat to an estrus female as an elicitation cue to evoke USV. The protocol has both manual [13] and semi-automated versions [14]. However, apart from an unpractical requirement of estrus females, there are documentation flaws (described in Discussion), and finally, it is focused mainly on the overall and unspecified increase in the number of USV emitted.

The goal of this study was to design a new model of operant conditioning in which rats would be rewarded for every 50-kHz USV emitted as well as to describe USV emitted for food reward. Such protocol could be the foundation for further more specific and complex USV-training to build on, e.g., rewarding rats for specific USV emissions. We set to test an array of parameters in the protocol such as the number of rewards or training days, and the addition of USV eliciting cues like USV-playback [see 15–17] and rat-scented cage bedding [comp. 18, 19] to optimize USV instrumental conditioning.

## Experimental procedures

### Animals and housing

All experiments were approved by the Second Local Ethical Committee in Warsaw. Naïve Wistar male rats (7–8 weeks of age, from The Center for Experimental Medicine of the Medical University of Bialystok, Poland) were housed in pairs. Standard chow and water were provided *ad libitum*, a 12 h light-dark cycle, and an ambient temperature of 22–25°C.

### Operant conditioning box

Experiments were conducted in a modular operant conditioning cage with a grid floor, pellet receptacle, and pellet dispenser (Behavioral Test Packages for Rat, Med Associates Inc., Fairfax, VT, USA) controlled by Med-PC® V software (version 5.08, Med Associates Inc.). Additionally, there was a USV condenser microphone CM16/CMPA (UltraSoundGate, Avisoft

Bioacoustics, Glienicke/Nordbahn, Germany) placed in the glass ceiling of the cage and an ultrasonic speaker (Vifa, Avisoft Bioacoustics) placed in the corner of the cage, next to the pellet receptacle, connected to an UltraSoundGate Player 116 (Avisoft Bioacoustics). Both USV-playback and recording were performed using Avisoft Recorder USGH software (version 1.0.0.1, Avisoft Bioacoustics). The locomotor activity was recorded with a FLIR® camera (Teledyne FLIR LCC, Wilsonville, OR, USA) mounted behind or above the cage and controlled via Spinnaker® SDK software (version 1.15.0.63, Teledyne FLIR LCC, Wilsonville, OR, USA); home-cage activity was recorded with a Basler camera (acA1300-60gc, Basler AG, Ahrensburg, Germany) controlled via EthoVision XT software (version 10, Noldus, Wageningen, Netherlands). Non-flavored sucrose pellets weighing 45 mg (TestDiet®, St. Louis, MO, USA) were used as rewards in conditioning. The cage was cleaned and thoroughly wiped using 10% EtOH between animals.

## Experimental schemes: USV-training

Habituation [based on 15, 16]: Rats were habituated to the new facilities after arrival (for 7 days), then to the experimenter (ADW; rats were handled once per day for 2 min for 5 days) and sucrose pellets. Ten sucrose pellets were given once per day for 3 days to each rat in their home cages to prevent food neophobia during the training phase. Animals were food restricted for 3 days before conditioning [20]; and received the amount of rat chow per day to maintain 90 or 95% of body weight under *ad libitum* access to food. To monitor the amount of feed consumed, the animals were singly housed (Fig 1).

Training phase [based on 20, 21]: Rats were individually placed once per day (for 7, 10 or 14 days) into an operant box; and training started after 10 s at which each rat received a reward for every 50-kHz USV emitted after consumption of the previous food reward (a sucrose pellet) and only when its nose was out of the pellet receptacle (protocols 1–6, EXP n = 76, Table 1). The operant conditioning was performed manually by a dedicated researcher (ADW). When observing/seeing on the monitor screen and hearing through headphones a USV, the pellet dispenser would immediately be activated by pressing a keyboard key. To determine the time from the rat USV emission to the pellet delivery, 10 random intervals were calculated from 10 randomly chosen rats. There was a 552.0 ± 46.2 ms interval between the start of the rewarded USV and the sound of the pellet dispenser; and a 521.3 ± 48.0 ms interval between the end of the rewarded USV and the pellet dispenser sound (S1 Fig). We used Ultra-SoundGate 116Hb (Avisoft Bioacoustics) with a stereo 3.5 mm mini jack connector to acoustically monitor the incoming ultrasounds through headphones; an undersampling technique was used to convert the USV into audible signals. In some protocols, two elicitation cues were used, i.e., bedding with the smell of a familiar rat and USV played from the speaker. The training ended when the rat reached the maximum number of rewards (10 or 30) or after 15 min.

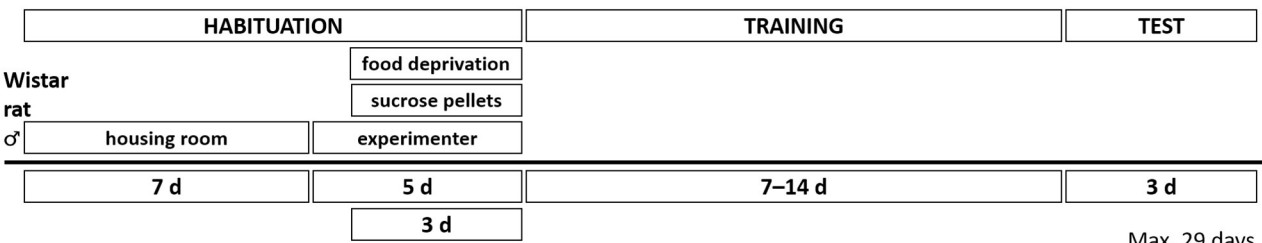

**Fig 1. General scheme of the experiment.** Each of the experiments consisted of three phases: habituation, training, and test.

**Table 1. Summary of experimental protocols (listed under #); EXP–conditioned rats; CTL–control rats.**

| # | Number of rats | | | Food deprivation [%] | Habituation to the box | Phase of the day | Maximum number of rewards | Number of sessions | | Elicitation cues |
|---|------|-----|-----|----|-----|-------|----|----------|------|------------------|
|   | All | EXP | CTL |    |     |       |    | training | test |                  |
| 1 | 16 | 8  | 8  | 95 | no  | light | 10 | 10 | 3 | none |
| 2 | 16 | 8  | 8  | 90 | no  | light | 10 | 14 | 3 | none |
| 3 | 20 | 10 | 10 | 90 | no  | light | 10 | 14 | 3 | bedding, playback |
| 4 | 20 | 10 | 10 | 90 | no  | light | 30 | 14 | 3 | bedding, playback |
| 5 | 20 | 20 | -  | 90 | no  | dark  | 30 | 14 | 3 | bedding, playback |
| 6 | 20 | 20 | -  | 90 | yes | dark  | 30 | 7  | 3 | none |

Test phase: Rats were placed individually, once per day for 3 days into an operant box for 5 min. The rats did not receive any rewards (Fig 1, Table 1).

In the case of Protocol 6, rats were divided into two groups: conditioned in the conditioning box (n = 10) and a home cage (n = 10)–the same as used previously [15, 16] during playback experiments. The home cage was adapted to conduct conditioning (installed with pellet receptacle, pellet dispenser, microphone, and camera, Fig 2B). Habituation to the conditioning environment (conditioning box or home cage) was conducted once per day (3 min session) for 4 days, followed by training and test sessions. For home cage habituation, we wanted to replicate the effect observed in playback experiments in which rats vocalized after being placed in the same familiar conditions for the 5th time [17] to rule out whether is it the lack of habituation causing stress/novelty effect and thus a smaller number of USV emissions compared to previous studies. Finally, we did not observe any major improvement in conditioning with more habituation.

## Playback USV used as elicitation cues

The USV playbacks were recordings from our previous experiments [16, 17] cleaned from background noise, extracted, and arranged. These recordings (50 ms or 3 s) were played when the rat did not vocalize for 90 s during the training session. If after playing the recording, the rat still did not vocalize for another 90 s, another USV set was played. There was no limit to the number of playbacks during a training session. Four different USV playback sets were used. In protocols 3–4, sets 1–3 were presented in succession; in protocol 5, sets 1, 2 and 4 were presented in succession. The properties of each USV set are presented below:

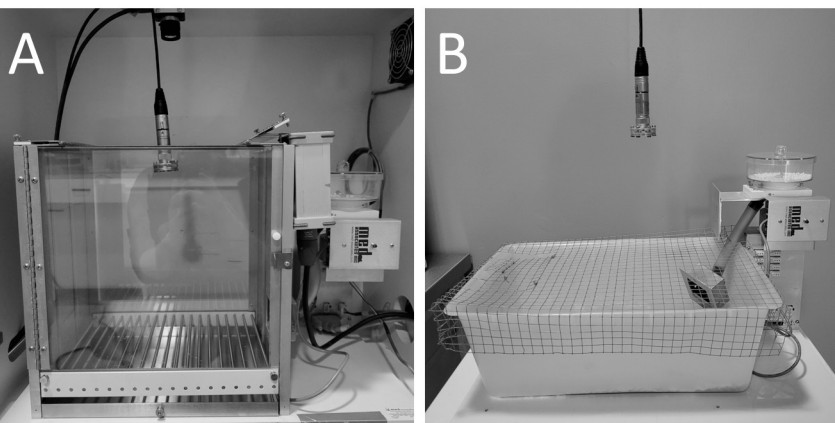

**Fig 2. Experiments were conducted in the operant conditioning box (A, protocols 1–6) or a home cage (B, protocol 6).**

1. USV modulated, 15 USV calls (average duration 53.6 ± 2.7 ms; average peak frequency 68.3 ± 1.6 kHz; see below for peak frequency definition) total duration 3 s;

2. USV non-modulated, 15 USV calls (average duration 68.3 ± 7.3 ms; average peak frequency 53.6 ± 0.8 kHz), total duration 3 s;

3. USV modulated, 1 USV (duration 41.4 ms, peak frequency 59.4 kHz), total duration 50 ms;

4. USV modulated, 24 USV calls (average duration 29.2 ± 3.6 ms; average peak frequency 58.9 ± 1.3 kHz), total duration 3 s; these USV were obtained from a rat exploring a cage with the scent of another rat.

### Experimental schemes: Control protocols, nosepoke-training

As a control to the experiments with USV as a conditioned response, rats were trained to poke their nose into the nosepoke hole; EXP = 36. The habituation phase was shortened as these nosepoke experiments were conducted after the USV-training (protocols 2, 5, and 6). The animals underwent food restriction to 90% of initial body weight. Followed by 5 days of training (one session per day) and each session ended either when the rat reached the maximum number of rewards or after 15 min. Lastly, 3 test sessions were conducted once per day, for 5 min; during which rats did not receive any rewards.

### Audio analysis

Audio files (.wav) were analyzed using SASLab Pro (Avisoft Bioacoustics, Glienicke/Nordbahn, Germany). Recordings were transferred into high-resolution spectrograms (frequency resolution: 391 Hz; time resolution: 0.64 ms) by conducting Fast Fourier Transform (512 FFT-length, 100% frame, FlatTop window, and 75% time window overlap). The ultrasounds were marked manually, the following values were obtained: duration, amplitude (average of all instantaneous spectra within a USV; reported as "amplitude" or "intensity" in the text), and peak frequency (highest power peak in the averaged spectrum of the entire element; regarded as "frequency" throughout the text). The detected USV was assigned to one of four categories: 50-kHz (frequency >32 kHz, duration <150 ms), short 22-kHz (frequency of 18–32 kHz, duration <0.3 s), long 22-kHz (frequency of 18–32 kHz, duration >0.3 s), 44-kHz USV (frequency >32 kHz, duration >150 ms).

### Statistical analysis

Figures and statistical analysis were performed in GraphPad Prism software (version 8.4.3, GraphPad Software, San Diego, CA, USA). The figures show means with standard errors of the mean (± SEM). All data were analyzed using non-parametric tests: two-tailed Wilcoxon, two-tailed Mann-Whitney, and Friedman. The Friedman test was used to assess a few-days tendency, while the Wilcoxon test was used for the difference between the first and the last day. The threshold of significance was $p < 0.05$. Results of statistical analyses are presented in the S2–S9 Tables.

## Results

### A subset of rats vocalizes for reward; all rats divided into 6 groups

In summary, 20% of rats (n = 15/76, Table 2, S1 Table) showed changes in behavior that demonstrated learning to vocalize for a reward ("potentially learning" rats, PL), i.e., they obtained the maximum number of rewards during each of the 3 last training sessions (at minimum).

**Table 2. Groups and subgroups of rats determined by the number of rewards achieved.**

| Group | Group description | Number of rats in group | | | |
|---|---|---|---|---|---|
| | | All | By number of training sessions | | |
| | | | 7 | 10 | 14 |
| **PL-SUM** | all potentially learning rats; achieved the maximum number of rewards during all last 3 training sessions | **15** | **5** | **2** | **8** |
| PL-MAX | potentially learning; ("maximum"; achieved the maximum number of rewards in all training sessions) | 5 | 3 | 0 | 2 |
| PL-PROG | potentially learning; ("progress"; achieved the maximum number of rewards in all the last 3 training sessions, but not in all training sessions) | 10 | 2 | 2 | 6 |
| **NL-SUM** | all non-learning rats; did not achieve the maximum number of rewards during all 3 last training sessions | **61** | **15** | **6** | **40** |
| NL-D1 | non-learning; ("day 1"; achieved the maximum number of rewards in the first training session, but did not obtain them in any of the last 3 training sessions) | 17 | 5 | 1 | 11 |
| NL-SGL | non-learning; ("single"; achieved the maximum number of rewards only in one training session, but not the first one) | 5 | 0 | 1 | 4 |
| NL-CEN | non-learning; ("center"; achieved the maximum number of rewards in a minimum of two consecutive training sessions, but not the first two or last two) | 5 | 0 | 0 | 5 |
| NL-0 | non-learning; ("zero"; did not achieve the maximum number of rewards in any training session) | 34 | 10 | 4 | 20 |
| NL-SUM/0 | non-learning; (rats from the NL groups excluding the NL-0 group) | 27 | 5 | 2 | 20 |

Also, the majority of the rats (80%, n = 61/76, Table 2, S1 Table) did not learn to vocalize for reward ("non-learning" rats, NL), i.e., these rats did not reach the maximum number of rewards during the last 3 training sessions. Therefore, in general, rats did not show significant increases in USV emissions during training regardless of the protocol modifications employed (S2a–S2g Fig, S2a and S2b Table). Simultaneously, these rats, when tested in nosepoke-training, showed learning (S2j–S2l Fig, S2c–S2e Table). For more detailed analysis, several subgroups were differentiated among NL and PL rats (Table 2, S1 Table, comp. Fig 3A–3F).

## Potentially learning rats showed higher call rate than non-learning rats

Call rate (number of USV emitted per 1 min) was higher in PL-SUM rats than in NL-SUM rats in training sessions 2–14 (Mann-Whitney, Fig 3H, S3b Table). In PL-SUM rats, the call rate remained constant over the first seven days of training, while in NL-SUM rats, it decreased (days 1–7: PL-SUM, $p = 0.1426$, Friedman; $p = 0.1645$, Wilcoxon; NL-SUM, $p < 0.0001$, Friedman; $p < 0.0001$, Wilcoxon; Fig 3H, S3a Table). The call rates of PL-SUM vs. NL-SUM rats also differed in the test sessions ($p < 0.0001$ for each day, Mann-Whitney, Fig 3H, S3b Table). After dividing PL-SUM and NL-SUM rats into subgroups (Table 2), a decrease in the call rate was observed in NL-D1 (days 1–7: $p = 0.0004$, Friedman; $p = 0.0039$, Wilcoxon) and NL-0 rats (days 1–7: $p < 0.0001$, Friedman; $p < 0.0001$, Wilcoxon, Fig 3G, S3a Table). Finally, the increase in call rate between the last day of training and the first test session was higher in PL-SUM rats ($14.5 \pm 4.8$) than in NL-SUM rats ($5.9 \pm 1.7$, Fig 3H), though not significantly different ($p = 0.0537$, Mann-Whitney). However, the difference was significant after removing one outlier, through Tukey's fences method [22], in NL-SUM rats in which the increase in call rate changed to $4.5 \pm 1.0$ ($p = 0.0326$, Mann-Whitney).

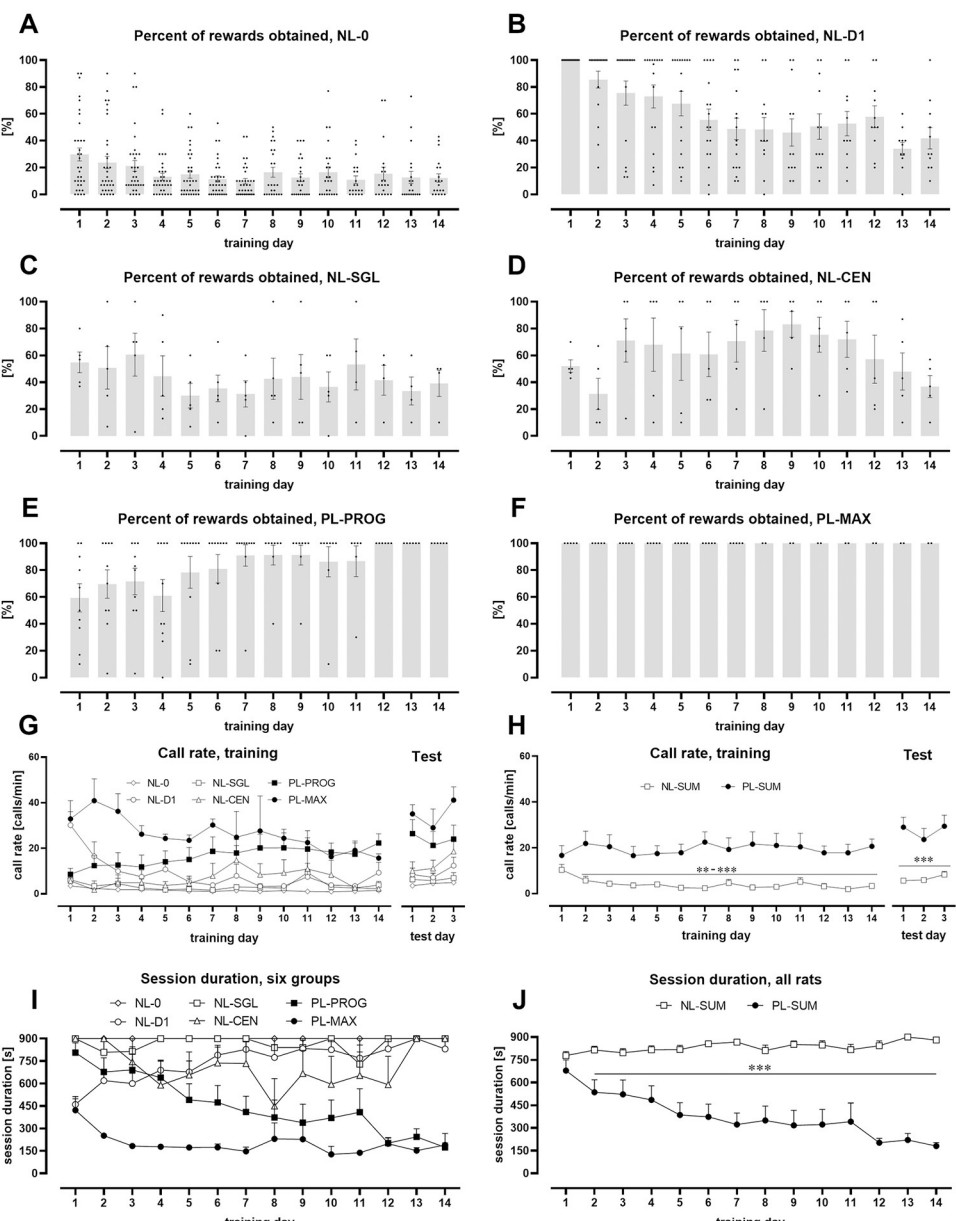

**Fig 3. Percentage of rewards obtained (A-F), call rate (GH), and session duration (IJ) in rats grouped by performance in USV-training. A.** NL-0: Rats that did not obtain the maximum number of rewards in any training session (NL-0, non-learning "zero", n = 34). **B.** NL-D1: Rats that obtained the maximum number of rewards on day 1 of training but not in the last 3 training sessions (NL-D1, non-learning "day one", n = 17). **C.** NL-SGL: Rats that obtained the maximum number of rewards in only one training session, but not the first (NL-SGL, non-learning "single", n = 5). **D.** NL-CEN: Rats that obtained the maximum number of rewards in a minimum of two consecutive training sessions, but not in the first two or last two (NL-CEN, non-learning" center", n = 5). **E.** PL-PROG: Rats that obtained the maximum number of rewards during each of the last 3 training sessions, but not during other training sessions (PL-PROG, potentially learning "progress", n = 10). **F.** PL-MAX: Rats that obtained the maximum number of rewards in all training sessions (PL-MAX, potentially learning "maximum", n = 5). **G.** Call rate in rats from experiments with 7, 10, or 14 training sessions divided into six groups. **H**. Call rate in rats from experiments with 7, 10, or 14 training sessions divided into two groups: NL-SUM (all non-learning rats,) and PL-SUM (all potentially learning rats). Note that the call rate of PL-SUM rats is higher than that of NL-SUM rats in almost all training and test sessions. **I.** Session duration in all rats divided into six groups (as in G). **J.** Duration of a training session in PL-SUM and NL-SUM rats. Session time was shorter in PL-SUM rats than in NL-SUM rats from the second training session onward. The bars and line plots represent the mean ± SEM. The dots represent individual values for each rat. **p < 0.01, ***p < 0.001, Mann-Whitney. Number of animals: A-F: NL-0, n = 20–34; NL-D1, n = 11–17; NL-SGL,

n = 4–5; NL-CEN, n = 5; PL-PROG, n = 6–10; PL-MAX, n = 2–5; G, training: NL-0, n = 8–22; NL-D1, n = 3–9; NL-SGL, n = 2–3; NL-CEN, n = 3–5; PL-PROG, n = 6–10; PL-MAX, n = 2–5; test: NL-0, n = 34; NL-D1, n = 17; NL-SGL, n = 5; NL-CEN, n = 5; PL-PROG, n = 10; PL-MAX, n = 5; H, training: PL-SUM, n = 8–15; NL-SUM, n = 16–39; test: PL-SUM, n = 15; NL-SUM, n = 61; I: NL-0, n = 20–34; NL-D1, n = 11–17; NL-SGL, n = 4–5; NL-CEN, n = 5; PL-PROG, n = 6–10; PL-MAX, n = 2–5; J: PL-SUM, n = 8–15; NL-SUM, n = 40–61. For statistical analysis see S3 and S4 Tables.

## Training sessions became gradually shorter in potentially learning rats

In PL-SUM rats, the duration of training sessions gradually decreased (rats with 7 training sessions, days 1–7, n = 5, p = 0.0154, Friedman, p = 0.0078, Wilcoxon; rats with 14 training sessions, days 1–14, n = 8, p = 0.0002, Friedman, p = 0.0078, Wilcoxon; all PL-SUM rats, days 1–7, n = 15, p < 0.0001, Friedman, p = 0.0002, Wilcoxon, Fig 3J, S4a Table), in contrast to NL-SUM rats in which the session time did not change (rats with 7 training sessions, days 1–7, p = 0.2267, Friedman, p = 0.0625, Wilcoxon; rats with 10 training sessions, days 1–10, p = 0.6114, Friedman, p > 0.9999, Wilcoxon; not shown) or rather increased (rats with 14 training sessions, days 1–14, n = 40, p = 0.0007, Friedman, p = 0.0029, Wilcoxon, not shown, S4a Table). Notably, for all NL-SUM rats, the session time increased from 778 ± 27 s (day 1) to 867 ± 19 s (day 7), n = 61, p = 0.0043, Friedman, p = 0.0053, Wilcoxon (Fig 3J, S4a Table). As a result, the duration of the training session was shorter in PL-SUM rats than in NL-SUM rats throughout sessions 2–14 (<0.0001–0.0014 p values, Mann-Whitney, Fig 3J, S4c Table). Importantly, there was a decrease in session duration in both PL-MAX rats (422 ± 92 s, day 1; 147 ± 10 s, day 7; n = 5, p = 0.0011, Friedman, p = 0.0625, Wilcoxon) and PL-PROG rats (807 ± 65 s, day 1; 410 ± 105 s, day 7; n = 10, p = 0.0003, Friedman, p = 0.0078, Wilcoxon; Fig 3I, S4b Table). For PL-PROG rats, the decrease in training duration was also observed for the 3 first days with the maximum number of rewards obtained (362 ± 66 s, day 1; 198 ± 19 s, day 3; n = 10, p = 0.0665, Friedman; p = 0.0195, Wilcoxon, S4b Table).

## Potentially learning rats showed increasing percentage of rewarded 50-kHz USV

Only results from experiments 4–6 (30 rewards) are analyzed in this section and the following sections. USV emitted were first assigned to one of four categories (see Introduction). Of those, 44-kHz USV [5] were sporadic and were not analyzed, i.e., a total of 34,894 >32 kHz USV were observed, of which only 102 (0.2%) were classified as 44-kHz USV. Similarly, long 22-kHz USV were rare and not analyzed i.e., in total of 4569 <32 kHz USV observed, only 229 (5.0%) were classified as long 22-kHz USV. The percentage of rewarded USV in all USV emitted during training increased gradually in PL-SUM rats over the 7-training sessions protocol (p = 0.0255, Friedman, p = 0.0625, Wilcoxon, Fig 4A, S5a Table) and 14 training days (p < 0.0001, Friedman; p = 0.0313, Wilcoxon, Fig 4B, S5b Table). Consequently, PL-SUM rats with 14 training sessions differed significantly from the three other groups analyzed, i.e., NL-SUM, NL-SUM/0, and NL-0 in the percent of rewarded USV in the last 3 training sessions (p = 0.0007–0.0426, Mann-Whitney, Fig 4B, S5d Table).

Analogously, the percent of rewarded USV in all 50-kHz USV emitted during training also increased in PL-SUM rats in the 7-training sessions protocol (p = 0.0059; Friedman, p = 0.0625, Wilcoxon, Fig 4C, S5a Table) and 14 training days (p < 0.0001, Friedman; p = 0.0313, Wilcoxon, Fig 4D, S5b Table). Consequently, PL-SUM rats with 14 training sessions differed from the NL-SUM (p = 0.0003) and NL-0 (p = 0.0007) groups in the last 3 training sessions (both Mann-Whitney, averages for the 3 days analyzed, Fig 4D, S5d Table).

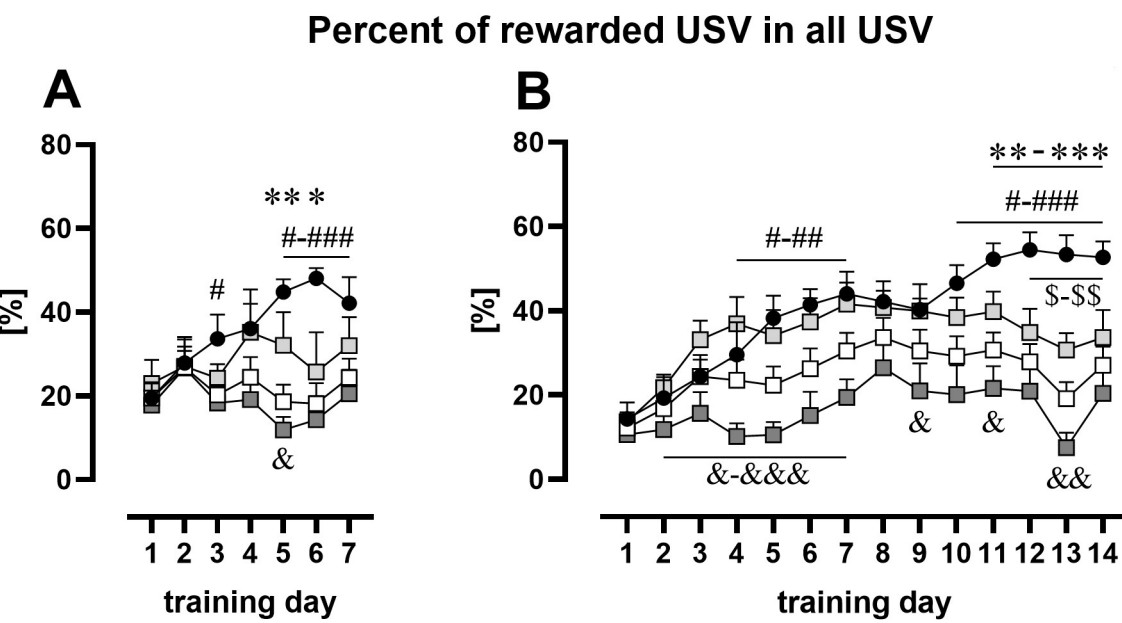

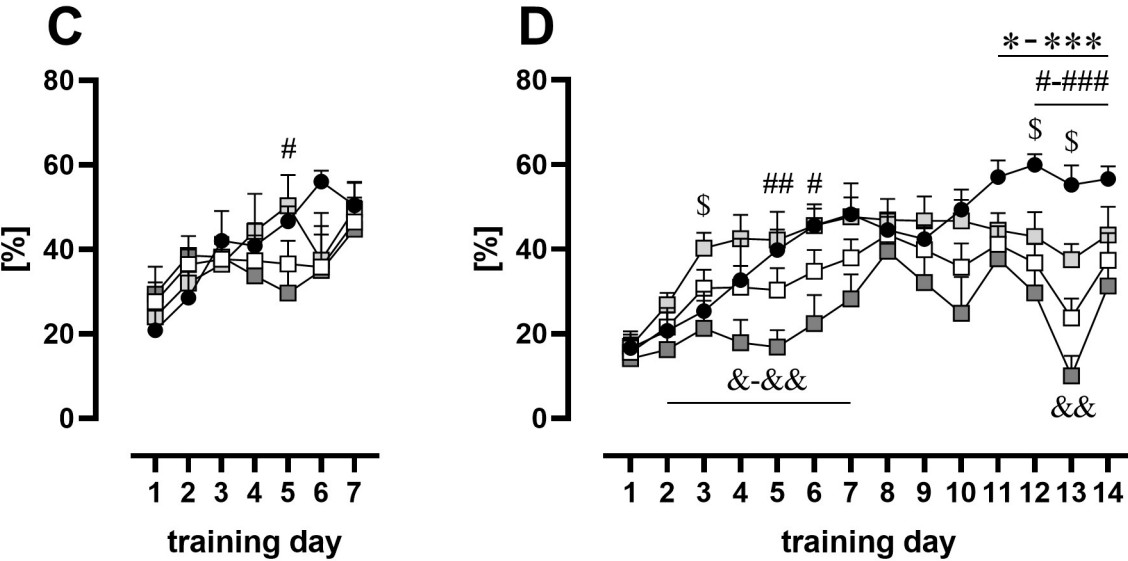

**Fig 4. Percentage of rewarded USV from the total number of USV (AB) and the number of 50-kHz USV (CD) during training in rats with 7 (AC) or 14 training sessions (BD). A.** Percentage of rewarded USV in all USV in 7-days-trained rats. **B.** Percentage of rewarded USV in all USV in 14-days-trained. **C.** Percentage of rewarded USV in 50-kHz USV in 7-days-trained rats. **D.** Percentage of rewarded USV in 50-kHz USV in 14-days-trained rats. Values for NL rats are presented in two ways: all NL rats are depicted as squares (NL-SUM), additionally, the NL-SUM group is divided into NL-0 rats (dark gray squares) that never reached the maximum number of rewards and NL-SUM/0 rats (NL-SUM rats excluding NL-0 rats, light gray squares). In PL-SUM rats, the percent of rewarded USV in all emitted USV and in 50-kHz USV increased in rats with 7 training sessions and 14 training sessions. Line plots represent mean ± SEM. * PL-SUM vs. NL-SUM, # PL-SUM vs. NL-0, $ PL-SUM vs. NL-SUM/0, & NL-SUM/0 vs. NL-0; one character (*, #, $ or &) p < 0.05, two p < 0.01, three p < 0.001; Mann-Whitney. Number of animals; AC: PL-SUM, n = 5; NL-SUM, n = 14–15; NL-SUM/ 0, n = 5; NL-0, n = 9–10; BD: PL-SUM, n = 6; NL-SUM, n = 15–16; NL-SUM/0, n = 8; NL-0, n = 7–8. For statistical analysis see S5 Table.

## More rewarded USV were emitted in series in potentially learning rats

This section pertains only to results from experiments 4–6, in which 30 rewards were used. Rewarded USV occurred in series, i.e. two or more consecutive USV rewarded. USV series occurred in both PL-SUM and NL-SUM/0 rats (NL-0 rats emitted too few USV). The number of USV emitted in series increased in PL-SUM rats with 7 training sessions (p = 0.0430, Friedman, p = 0.0625, Wilcoxon; S6a Table) and 14 training sessions (p = 0.0010, Friedman, p = 0.0625, Wilcoxon, S6a Table). In the majority of training sessions, PL-SUM rats emitted more USV in series than NL-SUM/0; it was significantly higher in some training sessions (p < 0.05, Mann-Whitney, S6b Table). Similarly, the maximum number of USV emitted in a single series was generally higher in the PL-SUM rats than in the NL-SUM/0, significantly higher in 3 training sessions (p < 0.01, Mann-Whitney, S6b Table). The overall duration of the USV emitted in series in PL-SUM or NL-SUM/0 rats did not change over training (S6c Table), but PL-SUM rats emitted significantly longer USV in series than NU-SUM/0 rats during numerous training sessions (p < 0.05, Mann-Whitney, S6d Table).

## Potentially learning rats exhibited increasing duration of 50-kHz USV

Only results from experiments 4–6 (30 rewards) are analyzed in this section. The duration of all 50-kHz USV emitted during the experiment (7 and 14 days of training) increased in all PL-SUM rats analyzed together during the first 7 days of training (p = 0.0092, Friedman, p = 0.0205, Wilcoxon, Fig 5A and 5C; S7a Table). Consequently, the 50-kHz USV of PL-SUM rats were longer than in analyzed NL groups during most of the sessions (Fig 5A and 5C, S7c and S7d Table). As a result, the average duration of 50-kHz USV of PL-SUM rats (29.9 ± 1.5 ms) was longer than in NL-SUM/0 (22.0 ± 1.2 ms, p = 0.0011), NL-SUM (19.6 ± 0.8 ms, p < 0.0001), and NL-0 rats (17.9 ± 0.7 ms, p < 0.0001; all Mann-Whitney). The 50-kHz USV of PL-SUM rats were also significantly longer during the test sessions (Fig 5B and 5D, S7c and S7d Table).

## Potentially learning rats exhibited increasing amplitude of 50-kHz USV

Only results from experiments 4–6 (30 rewards) are analyzed in this section; only NL-SUM/0 rats were analyzed as a control group. The amplitude of 50-kHz USV emitted during the experiment with 7 trainings increased in PL-SUM rats (p = 0.0224, Friedman, Fig 5E; S8a Table). Though the amplitude of 50-kHz USV did not change over training and test sessions in experiment with 14 trainings (S8a and S8b Table), the amplitude in PL-SUM rats was significantly higher than in NL-SUM/0 rats–in the last three training sessions and first test session (p < 0.05, Mann-Whitney, Fig 5G and 5H, S8d Table).

## Non-learning rats emitted more short 22-kHz USV than potentially learning rats

Again, only the results from experiments 4–6 (30 rewards) are analyzed in this section. In rats with 7 days of training, the percentage of short 22-kHz USV in the total of USV emitted did not change during the training in PL-SUM rats (training, p = 0.9190, Friedman, p = 0.8125, Wilcoxon; habituation and training, p = 0.9092, Friedman, p > 0.9999, Wilcoxon), but it increased in NL groups (habituation and training, NU-SUM/0, p = 0.0218, Friedman, p = 0.1250, Wilcoxon; NU-0, p < 0.0001, Friedman, p = 0.0098, Wilcoxon; NU-SUM, p < 0.0001, Friedman, p = 0.0006, Wilcoxon; Fig 6A, S9a Table). While, in rats with 14 days of training, this percentage significantly decreased in the PL-SUM rats (p = 0.0197, Friedman, p = 0.0625, Wilcoxon) and did not change in the NL rats (p = 0.2390–0.5130, Friedman,

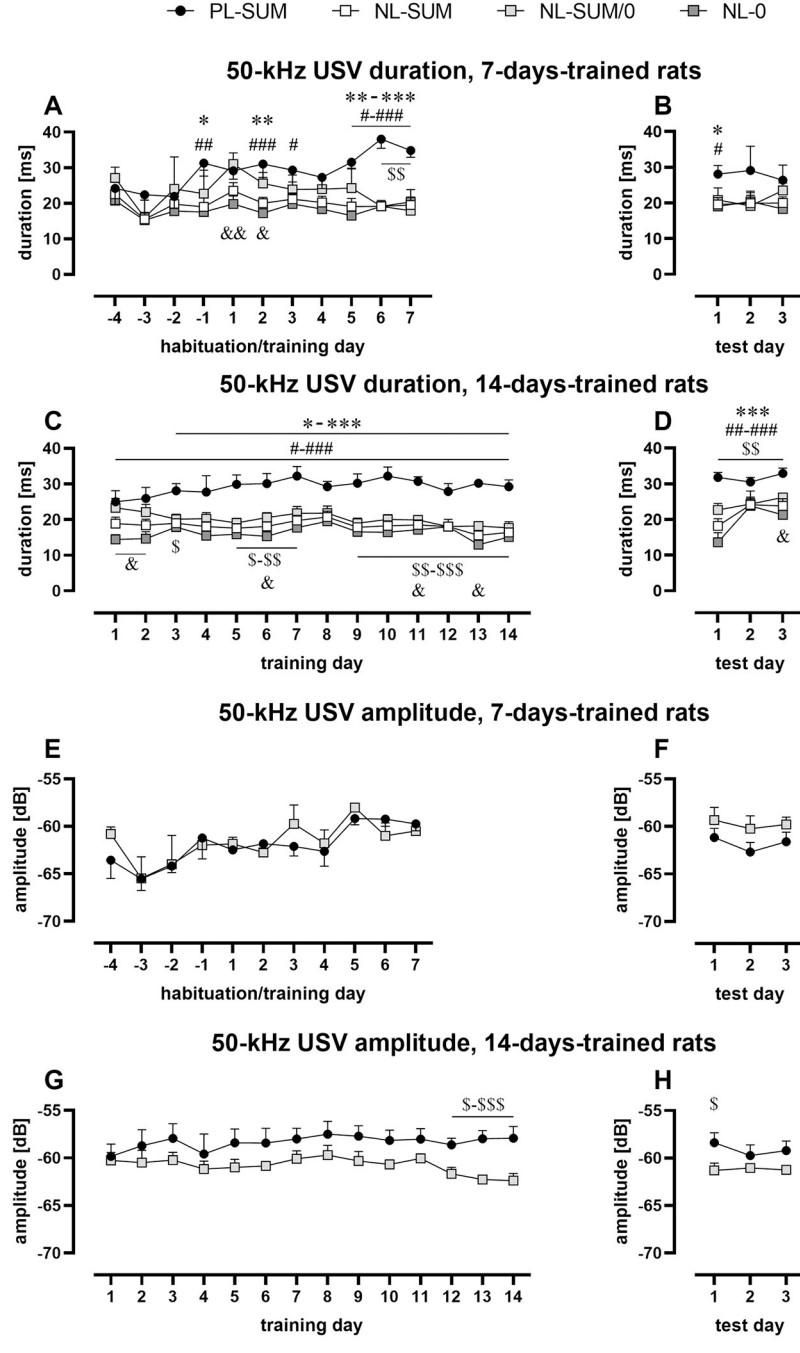

**Fig 5. Changes in 50-kHz USV duration (A-D) and amplitude (E-H) in rats with 7 (ABEF) or 14 (CDGH) training days during training (ACEG) and test (BDFH) sessions. A.** 50-kHz USV duration in 7-days-trained rats during habituation and training. **B.** 50-kHz USV duration during test sessions in 7-days-trained rats. **C.** 50-kHz USV duration in 14-days-trained rats. **D.** 50-kHz duration in 14-days-trained rats during test sessions. Values for non-learning rats are presented in two ways: all non-learning rats are depicted as white squares (NL-SUM), additionally, this group is divided into NL-0 rats (dark gray squares) that never received the maximum number of rewards and NL-SUM/0 rats (light gray squares). In PL-SUM rats, the duration of 50-kHz USV increased and was longer than in NL groups. **E.** 50-kHz USV amplitude in 7-days-trained rats during habituation and training. **F.** 50-kHz USV amplitude during test sessions in 7-days-trained rats. **G.** 50-kHz USV amplitude in 14-days-trained rats. **H.** 50-kHz USV amplitude in 14-days-trained rats during test sessions. In 14-days-trained PL-SUM rats, the amplitude of 50-kHz USV was higher than in NL-SUM/0 rats. The line plots represent the mean ± SEM. * PL-SUM vs. NL-SUM, # PL-SUM vs. NL-0, $ PL-SUM vs. NL-SUM/0, & NL-SUM/0 vs. NL-0; one character (*, #, $ or &) $p < 0.05$, two $p < 0.01$, three $p < 0.001$; Mann-Whitney. Number of animals; AB: PL-SUM, n = 5; NL-SUM, n = 13–15; NL-SUM/0, n = 4–5; NL-0, n = 8–10;

CD: PL-SUM, n = 6; NL-SUM, n = 15–16; NL-SUM/0, n = 8; NL-0, n = 7–8; EF: PL-SUM, n = 5; NL-SUM/0, n = 4–5; GH: PL-SUM, n = 6; NL-SUM/0, n = 7–8. For statistical analysis see S7 and S8 Tables.

p = 0.4609–0.7422, Wilcoxon, Fig 6C, S9a Table). Altogether, the percentage of short 22-kHz USV in PL-SUM rats was lower than in NL groups analyzed, especially near the end of training (Fig 6A and 6C, S9b and S9c Table). Overall, the percentage of short 22-kHz USV in PL-SUM rats (6.15 ± 2.14%) was lower than in NL-SUM/0 (15.67 ± 2.99%, p = 0.0031), NL-SUM (29.90 ± 3.58%, p < 0.001), and NL-0 rats (40.19 ± 4.42%, p < 0.001; all Mann-Whitney) when calculated for the first 7 days of training. Finally, the short 22-kHz USV of PL-SUM rats were also significantly fewer during the test sessions (Fig 6B and 6D, S9d and S9e Table). Exemplary spectrograms of short 22-kHz USV emitted by PL and NL rats are presented in S3 Fig. Additionally, no differences were observed between PL and NL rats regarding frequency of 50-kHz USV and short 22-kHz USV emitted as well as the length of short 22-kHz USV.

## Discussion

### Learning rats had different USV repertoire

In our hands, some rats (20%, n = 15 of 76 trained, potentially learning rats, PL rats) did most likely learn to vocalize to receive a reward in operant training, which was confirmed by several observations apart from achieving the maximum number of rewards. PL rats had longer and louder 50-kHz USV and emitted fewer short 22-kHz USV than NL (non-learning) rats. In particular, PL rats trained for 7 and 14 days emitted longer 50-kHz USV than NL rats, and the difference persisted during the test sessions. Longer 50-kHz USV were also emitted by PL rats in a series of rewarded USV–when compared with NL rats.

There is a limited number of publications where changes or differences in USV duration or amplitude were reported. An increase in USV duration was observed by Shembel et al. in an USV-rewarded instrumental paradigm [23]. Since longer USV were observed in our PL group and rats trained by Shembel et al., it can be assumed that this change is related to operant training and reward reinforcement. Shembel et al. also showed morphological changes in the laryngeal neuromuscular junction accompanying the 4-week-long training. In our previous study [16], rats after fear conditioning emitted 50-kHz USV of higher peak frequency and longer duration than control rats in response to ultrasonic playback. However, other teams related to Shembel et al. group [24, 25] showed no effect of USV-training on USV duration.

Shembel et al. (2021) also observed an increase in USV intensity [23], however no such increase was observed by Lenell et al [25] from the same laboratory. In our hands, the observed changes in USV amplitude were more subtle than those concerning USV duration–which may point to methodological difficulties regarding measuring USV intensity. Recently, Gonzales-Palomares et al. [26] reported over time changes in 22-kHz USV amplitudes in an odor fear conditioning task.

During training and testing, NL rats emitted short 22-kHz USV more frequently than PL rats. The biological significance of short 22-kHz USV is unclear. It has been suggested that short 22-kHz USV express an internal negative emotional state, while long 22-kHz calls are emitted in the presence of external danger [27]. These short 22-kHz USV were observed in rats during cocaine self-administration at a dose that did not allow for satiety [28]. Rats have also been reported to emit short 22-kHz USV when touched by an unfamiliar experimenter [3], presented with a warning sound signal before an electrical impulse [29], and during exploration of a new environment [30]. It can be proposed that the short 22-kHz USV emitted by NL rats in our experiments may reflect a stressful negative emotional state from not receiving

## Percent of short 22-kHz USV in all USV, 7-days-trained rats

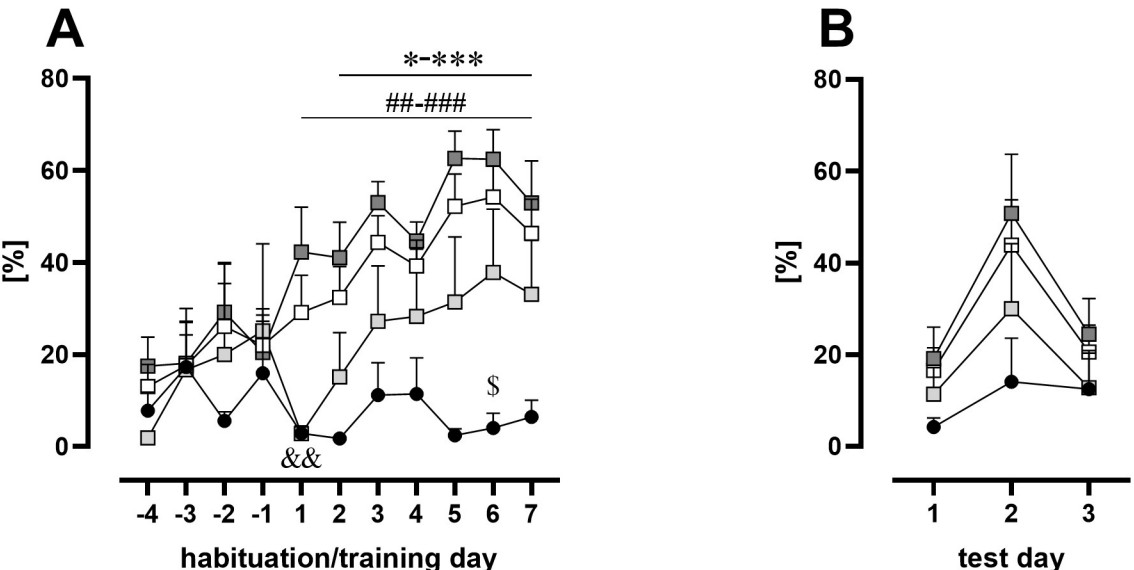

## Percent of short 22-kHz USV in all USV, 14-days-trained rats

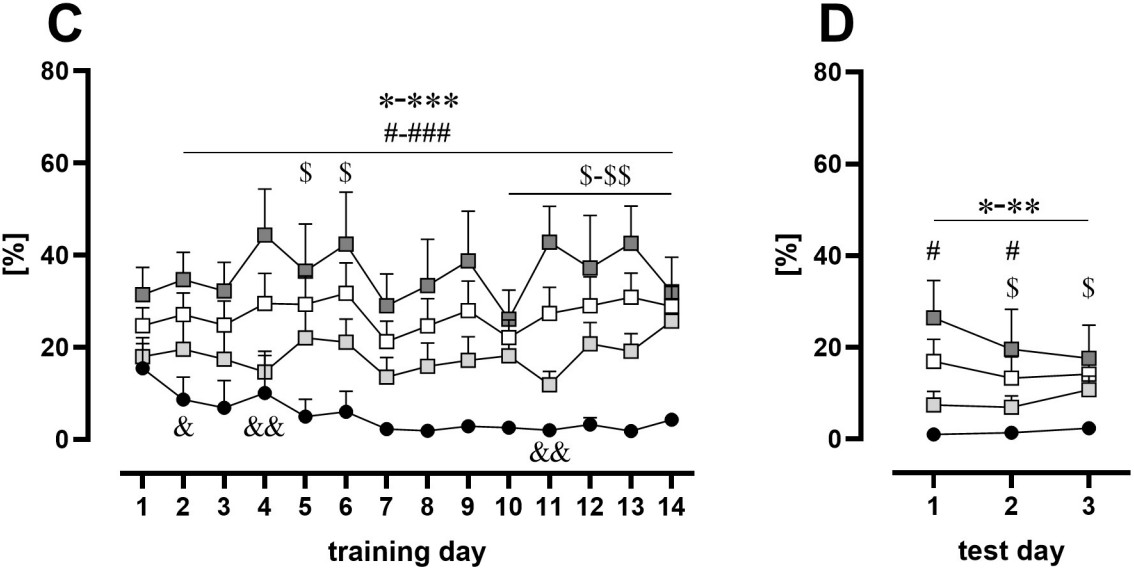

**Fig 6. Percentage of short 22-kHz USV in the total number of USV in rats with 7 (AB) or 14 (CD) training days during training (AC) and test (BD) sessions. A.** Percent of short 22-kHz USV in rats with habituation and 7 training sessions. **B.** Percent of short 22-kHz USV emitted by 7-days-trained rats in test sessions. **C.** Percent of short 22-kHz USV in rats with 14 training sessions. **D.** Percent of short 22-kHz USV emitted by 14-days-trained rats in test sessions. Overall, the percentage of 22-kHz USV was lower in PL-SUM rats in comparison with NL rats. The line plots (A-D) represent the mean ± SEM. * PL-SUM vs. NL-SUM; # PL-SUM vs. NL-0; $ PL-SUM vs. NL-SUM/0, & NL-SUM/0 vs. NL-0; one character (*, #, $ or &) p < 0.05, two p < 0.01, three p < 0.001; Mann-Whitney. Number of animals (AB): PL-SUM, n = 5; NL-SUM, n = 14–15; NL-SUM/0, n = 4–5; NL-0, n = 10; Number of animals (CD): PL-SUM, n = 6; NL-SUM, n = 16; NL-SUM/0, n = 8; NL-0, n = 8. For statistical analysis see S9 Table.

a reward and having to remain in the experimental cage for the maximum duration of the training session. The majority of USV-trained rats (n = 61 of 76, 80%, non-learning rats, NL rats) did not emit more 50-kHz USV over subsequent training sessions. Possible reasons are discussed later in the text below.

## Instrumental learning was confirmed by other observations

PL rats were selected based on arbitrary criteria, i.e., they all received a maximum set number of rewards during the last 3 training sessions. In summary, PL rats also exhibited:

- shortening of training sessions, contrary to the no change in duration observed in NL rats with even prolonging of sessions in some cases;

- a higher call rate during training and test sessions in comparison with NL rats;

- a gradual increase in the percentage of rewarded USV as well as in overall emitted 50-kHz USV (this suggests that rats were more purposeful in their ultrasonic emissions, e.g., with less USV emitted in the immediate proximity of food tray);

- a gradual increase in the number of rewarded USV emitted in series, the number was higher than in NL rats;

- longer and louder 50-kHz USV emitted in successive training sessions;

- a gradual decrease in the percentage of short 22-kHz USV.

The changes in the behavior of PL rats listed above could indicate an ongoing process of learning but could also be a result of dividing the animals using arbitrary criteria. However, additional observations, not pre-determined by the arbitrary criteria, supporting the hypothesis that PL rats did learn, included:

- shortening of training sessions in PL-MAX rats;

- shortening of training sessions during the first 3 training sessions with maximum rewards in PL-PROG rats;

- an increase in the number of USV between the last training session and the first test session, present in both PL and NL groups, though higher in the PL group.

## Potential reasons for difficulties in establishing vocalizations as operant response

There could be several reasons for the mere average 20% efficiency of our operant conditioning. Broadly speaking, the very nature of USV and their function could be the key factor–50-kHz USV play an important role in rat social communication to indicate an appetitive state [27, 31]. We could think of two problems associated with this. First, since 50-kHz USV serve mainly to express an emotional state, it may be difficult and unusual for rats to use them as a direct means to acquire a food reward. In contrast, lever pressing and especially nosepoking are motor and exploratory actions naturally used for food foraging and are thus more easily acquired as an operant behavior in exchange for a food reward. Second, we could have observed "false positives"; a rat could vocalize as a result of obtaining the reward, anticipating the reward [comp. 32, 33], and/or due to forming a positive association between the cage and the reward, i.e., rewarding food-restricted rats could produce an appetitive emotional state, resulting in a general increase in 50-kHz USV emissions and ensuing further food rewards–all

without forming a direct association between USV emission and reward. In other words, PL-rats may be "always happy optimists" vocalizing accordingly and becoming even more appetitive in their mood when rewarded.

Furthermore, several more subtle problems could be mentioned. Rats differ in their call rate at baseline conditions [18, 34, 35], which was low in most of our rats. Rats with low call rates had fewer chances to form an association between their USV and reward. Nevertheless, excluding rats with low call rates would lead to rejecting a significant number of potential learners while rats with an initial high call rate could turn out to be non-learners, e.g., NL-D1 rats.

Additionally, manual conditioning is prone to human error. We manually rewarded 50-kHz USV which are very short–lasting 10–150 ms [27]. In operant conditioning, the time between a behavior and a reward is very important and should be as short as possible since even a delay of 1 s is sufficient to delay acquisition [36, 37]. The reward was manually delivered according to the experimenter's reaction time which may not have been quick (about 0.5 s on average, see *Experimental Procedures*) or consistent enough. The time between behavior and reward delivery could be shortened by using an automated training method.

Food deprivation is commonly used in conditioning to motivate animals to seek food rewards [10, 37, 38]. Schwarting et al. demonstrated that rats emitted 50-kHz USV in a new experimental cage as well as in a familiar one, regardless of being food-deprived or with food *ad libitum*. However, the call rate was higher in animals with food *ad libitum*, which suggests that food deprivation may in fact suppress 50-kHz USV emission [34]. Also, to monitor food restriction for each animal, rats previously housed in pairs were separated into individual cages and this social change may also affect 50-kHz USV emission. On one hand, the isolation of an adult rat may increase the number of 50-kHz USV emitted [e.g., 17, 39–41]. On the other hand, singly-housed male rats have an increased need for socialization with another individual compared to rats housed in groups [42], in turn, isolation may cause USV with a higher emotional load and need for socialization to predominate, while USV related to exploration, foraging, and food consumption may decline. Therefore, single-housing the rats could have also impeded the learning process.

USV elicitation cues, e.g., USV-playback, were to increase the number of 50-kHz USV emitted by rats. Rats were shown to emit 50-kHz USV in response to USV-playbacks [17, 35]. It has also been shown that the amount of dopamine released in the *nucleus accumbens* during the first 50-kHz USV-playback presentation, resembles that following a food reward, however, the response quickly decreases with subsequent presentations of the recording [43]. This could be the possible reason, why presenting the 50-kHz USV-playback was not able to continue to elicit USV responses in our long-term experiments. Similarly, we added the cage bedding from a littermate, as it has been shown that rats vocalize at 50 kHz in response to the scent of other rats [18]. However, these experiments were only 4 days long and it can be speculated that prolonged presentation of the same smell may lose its effectiveness in eliciting USV emission.

As a result of the above-mentioned general difficulties, arguably, there are only a few USV-based operant conditioning protocols. Some researchers have used USV as stimuli to respond to, to distinguish between [e.g., 10, 11, 44, 45] and finally as rewarded operant responses [13, 14]. Notably, lengthy training of weeks or even months was needed to produce acceptable results. It is worth noting that extensive training, spanning weeks or even months, was required to achieve such results.

## Analysis of current USV-training protocol

To our best knowledge, in the current literature there is only one functioning operant conditioning protocol that reinforces USV emitted by rats [e.g., 13, 14]. However, we encountered

numerous discrepancies when scrutinizing the publications using this protocol. Firstly, the studies were only performed on male rats that vocalized when introduced to females in estrus, which is an impractical precondition. Moreover, 20–50% of these rats did not vocalize and were excluded from the training procedures [14]. Finally, even after weeks of daily training, the effectiveness of the protocol remains difficult to assess.

Across many publications using the same operant conditioning protocol, the stated key changes in rat behavior supporting evidence of learning are not properly supported by statistical analysis. This makes it difficult to assess whether "the rate of vocalizations can be increased" [13]; "all trained rats progressively increased their performance each week during the vocal training" [13, 14, 25]; "USV production steadily increased", and "the semi-automated procedure resulted in a similar increase in the number of USVs produced over the first 4 weeks of training" [14] are actually statistically conclusive results. Also, several aspects of the protocol are only vaguely described and inconsistent. Deprivation is not fully defined and often only mentioned as "restricted access to water" [13] or "restricted access to food" [14], when in actuality, e.g., the rats solely consumed pellet rewards during training and their weight would drop below 80% baseline [14]. Water deprivation was used in some publications [25, 46], while in two other cases animals were given water and food *ad libitum* despite using sucrose/food rewards in training [23, 24]. Finally, in some experiments, a "pen click" was used as a conditioned stimulus alongside a food/water reward as positive reinforcement [13, 25, 46] to gradually transition to a "pen click" reward alone. However, this "pen click" reward was not used in other studies [14, 23, 24], nor can we find data supporting the effectiveness of "pen clicks". Thus, it leaves a question about the necessity of using this tool in training and its effectiveness.

## Conclusion

In this study, rats that vocalized and received the reward demonstrated changes in their USV repertoire. Their 50-kHz USV became longer and louder while they emitted fewer short 22-kHz USV than the non-learning group. These rats, classified as potential learners, demonstrated some learning behaviors independent of the selection criteria. In general, the rats did not learn the USV-response as quickly as the nosepoke-response. USV may not easily become an operant response due to their biological role, i.e., communication of emotional state between conspecifics. To master this USV-dependent operant protocol, it may not be simply learning the association between emitting a 50-kHz call to receive a reward, but also learning to emit a biologically-deeply-rooted appetitive call under circumstances that do not elicit a positive emotional state. Finally, we raise some questions about the effectiveness and reliability of the only other currently used USV-training protocol developed to increase the number of emitted USV.

## Supporting information

**S1 Fig. Five exemplary spectrograms showing USV emission (rewarded behavior) followed by the release of a food reward.** Interval I–between the start of the rewarded USV and the sound of the pellet dispenser; Interval II–between the end of the rewarded USV and the sound of the pellet dispenser. First arrows point the USV, second arrows point the sounds of pellet dispenser and third arrows point the sounds of pellet's appearance in the pellet receptacle; see also *Experimental schemes*: *USV-training* in *Experimental Procedures*.
(TIF)

**S2 Fig.** Changes in the percentage of rewards rats obtained and the duration of training sessions in operant-conditioning protocols 1–6 with vocalization emissions (A-I) or nosepokes

(J-L) as rewarded responses. Rats could obtain a maximum of 10 or 30 rewards in one training session. A training session was terminated when a rat obtained the maximum number of rewards or the trial time exceeded 15 min. **A.** Protocol 1 (see also Fig 1 and Table 1 for protocols' description): Food-restricted (95% initial body weight) rat was trained for 10 sessions, with a maximum of 10 rewards. **B.** Protocol 2: Conditions from Protocol 1 were modified by increasing food deprivation to 90% and extending training to 14 sessions. **C.** Protocol 3: Conditions from Protocol 2 were modified by adding 2 vocalization-eliciting stimuli: bedding from a cagemate and USV playback. **D.** Protocol 4: Conditions from Protocol 3 were modified by increasing the maximum number of rewards to 30. **E.** Protocol 5: Conditions from Protocol 4 were modified by performing training sessions in the dark phase instead of the light phase. **F.** Protocol 6: Conditions from Protocol 5 were modified by using habituation to the experimental cage (4 sessions before training) and reducing the training to 7 sessions. **G.** Protocol 6: Conditions were the same as in F, with habituation and training in a cage identical to the home cage. In all protocols, there was no overall increase in the number of rewards obtained (A, B, D, E, F), rather a decrease was observed (C: $p < 0.0001$, Friedman, $p = 0.0020$, Wilcoxon; G: $p = 0.0143$, Friedman, $p = 0.0078$, Wilcoxon). **H.** Percent of rewards obtained during training from all protocols (1–6, A-G) pooled together; there was a decrease in the number of rewards (days 1–7: $p < 0.0089$, Friedman; $p = 0.0015$, Wilcoxon; days 1–14: $p < 0.0001$, Friedman; $p = 0.0022$, Wilcoxon, S2b Table). **I.** Duration of training sessions from all protocols (1–6, A-G) showed no change in time (days 1–7: $p = 0.7292$, Friedman; $p = 0.2324$, Wilcoxon; days 1–10: $p = 0.4746$, Friedman; $p = 0.7500$, Wilcoxon; days 1–14: $p = 0.9648$, Friedman; $p = 0.5412$, Wilcoxon, S2b Table). **J.** Percent of rewards obtained by rats in nosepoke-conditioning. The majority of rats, initially trained in protocols 2 (n = 8), 5 (n = 8), and 6 (n = 20) obtained 100% of the rewards in the first nosepoke-training session (27/28 in the 30-reward protocol, 8/8 in the 10-reward protocol), which was maintained to the end of training. **K.** Duration of the nosepoke-training sessions decreased for both 10 ($p < 0.0001$, Friedman, $p = 0.0078$, Wilcoxon) and 30 rewards training sessions ($p < 0.0001$, Friedman, $p < 0.0001$, Wilcoxon, S2c Table). **L.** Number of nosepokes during test sessions significantly decreased in subsequent test sessions for both maxima of 10 rewards ($p = 0.0099$, Friedman, $p = 0.0391$, Wilcoxon) and 30 rewards protocols ($p < 0.0001$, Friedman, $p < 0.0001$, Wilcoxon, S2d Table). The bars (A-H, J, L) and line plots (I, K) represent the mean ± SEM. The dots represent individual values for each rat. *$p < 0.05$, **$p < 0.01$, ***$p < 0.001$, Wilcoxon. Number of animals: A, n = 8; B, n = 8; C, n = 10; D, n = 10; E, n = 20; F, n = 10; G, n = 10; H, n = 76; I, 14 training sessions, n = 48, 10 sessions, n = 8, 7 sessions, n = 20; J-L, 10 rewards, n = 8; 30 rewards, n = 28. For statistical analysis see S2 Table.
(TIF)

**S3 Fig. Examples of short 22-kHz USV.** Emitted by non-learning rats (Nos. 1–3) and potentially learning rats (Nos. 4–6) during the first and last training sessions (first two columns) as well as the first and last test sessions (latter two columns).
(TIF)

**S1 Table. Percentage of rewards obtained by each rat during each training session.**
(PDF)

**S2 Table. Percentage of rewards obtained and duration of training sessions in rats trained with several protocols of instrumental learning with USV emissions or nosepokes as rewarded responses.**
(PDF)

**S3 Table. Call rate (USV/min) in PL and NL rats.**
(PDF)

**S4 Table. Session duration in NL and PL rats.**
(PDF)

**S5 Table. Percentage of rewarded USV in the total number of USV and total number of 50-kHz USV in rats during 7 and 14 training sessions.**
(PDF)

**S6 Table. Rewarded USV emitted in series: all rewarded USV emitted in series (number of USV), maximum number of USV in single series (max USV).**
(PDF)

**S7 Table. Duration of 50-kHz USV.**
(PDF)

**S8 Table. Amplitude of 50-kHz USV.**
(PDF)

**S9 Table. Percent of short 22-kHz USV in all USV.**
(PDF)

## Author Contributions

**Conceptualization:** Agnieszka D. Wardak, Robert K. Filipkowski.

**Data curation:** Agnieszka D. Wardak, Rafał Polowy.

**Formal analysis:** Agnieszka D. Wardak, Krzysztof H. Olszyński.

**Funding acquisition:** Robert K. Filipkowski.

**Investigation:** Agnieszka D. Wardak.

**Resources:** Jan Matysiak.

**Software:** Rafał Polowy.

**Supervision:** Robert K. Filipkowski.

**Writing – original draft:** Agnieszka D. Wardak, Krzysztof H. Olszyński, Robert K. Filipkowski.

**Writing – review & editing:** Agnieszka D. Wardak, Krzysztof H. Olszyński, Rafał Polowy, Jan Matysiak, Robert K. Filipkowski.

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
