## [Decision Letter · Decision Letter 0]

6 Nov 2023

PONE-D-23-31875Rats that learn to vocalize for food reward emit longer and louder appetitive calls and fewer short aversive callsPLOS ONE

Dear Dr. Filipkowski,

Thank you for submitting your manuscript to PLOS ONE. After careful consideration, we feel that it has merit but does not fully meet PLOS ONE’s publication criteria as it currently stands. Therefore, we invite you to submit a revised version of the manuscript that addresses the points raised during the review process.

ACADEMIC EDITOR: - please do address properly very serious concerns of the expert rev. We are doing our best to provide detailed and professional revision process and it usually requires time

We look forward to receiving your revised manuscript.

Kind regards,

Prof. Dr. Dragan Hrncic, MD, MSc, MBE, PhD

Academic Editor

PLOS ONE

Journal Requirements:

3. Thank you for stating the following financial disclosure: "This work was funded by the National Science Centre, Poland, grant OPUS no. 2015/19/B/NZ4/03393 and by Mossakowski Medical Research Institute Research Fund, grant no. FBW-17." 

Reviewers' comments:

Reviewer's Responses to Questions

**Comments to the Author**

1. Is the manuscript technically sound, and do the data support the conclusions?

Reviewer #1: Yes

2. Has the statistical analysis been performed appropriately and rigorously? 

Reviewer #1: Yes

3. Have the authors made all data underlying the findings in their manuscript fully available?

Reviewer #1: Yes

4. Is the manuscript presented in an intelligible fashion and written in standard English?

Reviewer #1: No

5. Review Comments to the Author

Reviewer #1: This MS reports the authors` attempts to condition 50-kHz calls of male Wistar rats using food pellets as rewards. The outcomes of their efforts were somehow disappointing because only about 20% of a rather large sample of subjects showed evidence for instrumental conditioning, and the authors undertook several control procedures in order to enhance the likelihood of conditioning, which were more or less ineffective. As an additional control, they trained several of their rats in a “conventional” procedure, i.e. nose-pokes for the same type of reward, which – not surprisingly – led to substantial conditioning. The authors present their results in great detail in sub-groups of learners and non-learners (based on post-hoc criteria), including call rates over training days and descriptive evidence for changes in call features. Also, they discuss possible reasons for their largely negative outcomes, especially that 50-kHz calls may not be as easy to be conditioned as “conventional” operants, like lever-pressing or nose-poking. Altogether, the MS touches a very interesting aspect of ultrasonic vocalizations and is clearly worth to be published. Before that, however, several aspects should be addressed by the authors:

Overall: The MS would still profit from further stilistic revision.

Abstract: lines 24-25 are somehow redundant with statements made just before

Introduction: lines 44-45, punishment decreases rather than increases behavior

Methods: Major point of criticism: The authors provide no details how conditioning was actually performed. Only later, one reads that an observer probably made a decision and activated pellet delivery, but these key issues of the MS should be clearly explained including statements how long the intervals between a given call and pellet delivery actually was. Spectrographic examples could help here, because they probably show both, call and machine/pellet sounds.

Results:

Did the authors test extinction effects in the USV experiments? One could expect that call rates go up early during extinction like "conventional" operants.

Do rats with higher baseline call rates call more/learn during conditioning?

Lines 255 to 257: Why do the 2 totals differ?

Lines 265-267 (and elsewhere): What is meant by share?

Lines 285 and following: What defines a series apart from “2 or more”? inter-call lengths?

Overall, I find the result sections partly too detailed, also because of a lot of descriptive and post-hoc “significant” results compared to those of confirmatory nature.

Also, the MS is sometimes difficult to read because the figure legends (without figures) repeatedly interrupt the text. This would of course not be the case in a publication, but makes the work harder for the reviewer.

Short 22-kHz calls: Please provide exemplary spectrograms.

Discussion:

Line 384: What is meant by “other teams”?

Lines 455-456: How long were the times?

6. PLOS authors have the option to publish the peer review history of their article (what does this mean?). If published, this will include your full peer review and any attached files.

Reviewer #1: No

---

## [Author Response · Author response to Decision Letter 0]

11 Dec 2023

ANSWERS TO REVIEVER’S QUESTIONS

Overall: The MS would still profit from further stilistic revision.

Answer: Before submission, the manuscript was corrected by a native speaker professionally involved in text revision within the field of neuroscience. It was again revised before submitting this time.

Abstract: lines 24-25 are somehow redundant with statements made just before.

These sentences may sound redundant but they convey two separate messages: i. potentially learning rats show changes during training, i.e., from day to day; ii. potentially learning rats differ from non-learning rats during training. We corrected these statements in the Abstract and expanded the second sentence. It now reads: “As a result, potentially learning rats, when compared to non-learning rats, displayed…” (line 23).

Introduction: lines 44-45, punishment decreases rather than increases behavior

We respectfully disagree. We are referring to a behavioral change. Punishment can affect/change behavior. However, we revised that statement in the Introduction and changed “to increase” into “to influence” – which, we hope, will be less confusing (line 44).

Methods: Major point of criticism: The authors provide no details how conditioning was actually performed. Only later, one reads that an observer probably made a decision and activated pellet delivery, but these key issues of the MS should be clearly explained including statements how long the intervals between a given call and pellet delivery actually was. Spectrographic examples could help here, because they probably show both, call and machine/pellet sounds.

Answer: We added the details regarding how conditioning was performed in Materials and Methods. Please see lines 108-117: “The operant conditioning was performed manually by a dedicated researcher (ADW). When observing/seeing on the monitor screen and hearing through headphones a USV, the pellet dispenser would immediately be activated by pressing a keyboard key. To determine the time from the rat USV emission to the pellet delivery, ten random intervals were calculated from 10 randomly chosen rats. There was a 552.0 ± 46.2 ms interval between the start of the rewarded USV and the sound of the pellet dispenser; and a 521.3 ± 48.0 ms interval between the end of the rewarded USV and the pellet dispenser sound (S1 Fig). We used UltraSoundGate 116Hb (Avisoft Bioacoustics) with a stereo 3.5 mm mini jack connector to acoustically monitor the incoming ultrasounds through headphones; an undersampling technique was used to convert the USV into audible signals.”

We also added a figure with spectrographic examples (S1 Fig).

Results: Did the authors test extinction effects in the USV experiments? One could expect that call rates go up early during extinction like "conventional" operants.

We tested extinction for 3 test days only. Indeed, we observed a higher call rate in potentially learning rats vs. non-learning rats (p < 0.0001, Mann-Whitney) for all 3 days. Also, the increase in call rate between training and test sessions was higher in potentially learning rats. 

Do rats with higher baseline call rates call more/learn during conditioning?

Answer: We defined “baseline call rate” as the call rate during the first training session and analyzed it in rats from protocols 5 and 6. Top 20% of rats with the highest baseline call rate belonged to either the PL-MAX (n = 3) or NU-D1 group (n = 5). Conversely, the lowest 20% of vocalizers were comprised of NL-0 rats (n = 9). Overall, we did not observe rats with high baseline call rates to usually call more/learn during conditioning. However, low vocalizers usually did not learn.

Lines 255 to 257: Why do the 2 totals differ?

Answer: The first total (”34,894 >32 kHz USV”) refers to all USV with frequencies above 32 kHz, which, according to literature, belong to the appetitive frequency band. The second total (“4569 <32 USV”) refers to all USV below 32 kHz, belonging to the aversive frequency band.

Lines 265-267 (and elsewhere): What is meant by share?

Answer: By the ”share of” we meant “the percentage of”. The text was changed (lines 276-279). 

Lines 285 and following: What defines a series apart from “2 or more”? inter-call lengths?

Answer: We defined a series of USV as 2 or more consecutive USV that were rewarded. We did not set a particular minimum inter-call length. 

Overall, I find the result sections partly too detailed, also because of a lot of descriptive and post-hoc “significant” results compared to those of confirmatory nature.

Answer: We tried to limit the results to only reporting significance to support key observations, however, we are open to reduce some of the results reported. 

Also, the MS is sometimes difficult to read because the figure legends (without figures) repeatedly interrupt the text. This would of course not be the case in a publication, but makes the work harder for the reviewer.

Answer: We are sorry for this inconvenience. We were following the manuscript submission guidelines: “Figure captions must be inserted in the text of the manuscript, immediately following the paragraph in which the figure is first cited (read order). Do not include captions as part of the figure files themselves or submit them in a separate document.”

Short 22-kHz calls: Please provide exemplary spectrograms.

Answer: We present exemplary spectrograms (S3 Fig) with examples of short 22-kHz USV emitted by non-learning rats and potentially learning rats during the first and last training sessions (first two columns) and first and last test sessions (latter two columns). 

Discussion: Line 384: What is meant by “other teams”?

Answer: By “other teams related to Shembel et al. group [25, 26]” we mean teams authoring the 25th and 26th referenced publications, i.e., “other teams” are Johnson et al. (2013) and Lenell et al. (2019) who reported no effect of USV-training on USV duration.

Lines 455-456: How long were the times?

Answer: It was defined before (see above, S1 Fig), see also lines 108-117. We also added, lines 466-467: “(about 0.5 s on average, see Experimental Procedures)”.

---

## [Decision Letter · Decision Letter 1]

2 Jan 2024

Rats that learn to vocalize for food reward emit longer and louder appetitive calls and fewer short aversive calls

PONE-D-23-31875R1

Dear Dr. Filipkowski,

We’re pleased to inform you that your manuscript has been judged scientifically suitable for publication and will be formally accepted for publication once it meets all outstanding technical requirements.

Kind regards,

Prof. Dr. Dragan Hrncic, MD, PhD

Academic Editor

PLOS ONE

Additional Editor Comments (optional):

Reviewers' comments:

Reviewer's Responses to Questions

**Comments to the Author**

1. If the authors have adequately addressed your comments raised in a previous round of review and you feel that this manuscript is now acceptable for publication, you may indicate that here to bypass the “Comments to the Author” section, enter your conflict of interest statement in the “Confidential to Editor” section, and submit your "Accept" recommendation.

Reviewer #1: All comments have been addressed

2. Is the manuscript technically sound, and do the data support the conclusions?

Reviewer #1: Yes

3. Has the statistical analysis been performed appropriately and rigorously? 

Reviewer #1: Yes

4. Have the authors made all data underlying the findings in their manuscript fully available?

Reviewer #1: Yes

5. Is the manuscript presented in an intelligible fashion and written in standard English?

Reviewer #1: Yes

6. Review Comments to the Author

Reviewer #1: none

nnnnnnnnnnnnnnnnnnnnnnnnnnnnnnnnnnnnnnnnnnnnnnnnnnnnnnnnnnnnnnnnnnnnnnnnnnnnnnnnnnnnnnnnnnnnnnnnnnnnnnnnnnnnnnnnnnnnnnnnnnnnnnnnnnnnnnnnnnnnnnnnnnnnnnnnnnnnnnnnnnnnnnnnnnnnnnnnnnnnnnnnnnnnnnnnnnnnnnnnnn

7. PLOS authors have the option to publish the peer review history of their article (what does this mean?). If published, this will include your full peer review and any attached files.

Reviewer #1: No

---

## [Editor Report · Acceptance letter]

31 Jan 2024

PONE-D-23-31875R1 

PLOS ONE

Dear Dr. Filipkowski, 

I'm pleased to inform you that your manuscript has been deemed suitable for publication in PLOS ONE. Congratulations! Your manuscript is now being handed over to our production team.

Kind regards, 

on behalf of

Professor Dragan Hrncic 

Academic Editor

PLOS ONE